# Evaluation of ROTARIX^®^ Booster Dose Vaccination at 9 Months for Safety and Enhanced Anti-Rotavirus Immunity in Zambian Children: A Randomised Controlled Trial

**DOI:** 10.3390/vaccines11020346

**Published:** 2023-02-03

**Authors:** Natasha Makabilo Laban, Samuel Bosomprah, Michelo Simuyandi, Mwelwa Chibuye, Adriace Chauwa, Masuzyo Chirwa-Chobe, Nsofwa Sukwa, Chikumbutso Chipeta, Rachel Velu, Katanekwa Njekwa, Cynthia Mubanga, Innocent Mwape, Martin Rhys Goodier, Roma Chilengi

**Affiliations:** 1Department of Infection Biology, Faculty of Infectious and Tropical Diseases, London School of Hygiene and Tropical Medicine, London WC1E 7HT, UK; 2Enteric Disease and Vaccine Research Unit, Centre for Infectious Disease Research in Zambia, Lusaka P.O. Box 34681, Zambia; 3Department of Biostatistics, School of Public Health, University of Ghana, Accra P.O. Box LG13, Ghana; 4Department of Global Health, Amsterdam Institute for Global Health and Development (AIGHD), Amsterdam University Medical Centers, University of Amsterdam, Paasheuvelweg 25, 1105 BP Amsterdam, The Netherlands; 5Department of Biomedical Sciences, School of Health Sciences, University of Zambia, Lusaka P.O. Box 50110, Zambia; 6Division of Medical Microbiology, Department of Pathology, Stellenbosch University & National Health Laboratory Service, Tygerberg Hospital Francie van Zijl Drive, Tygerberg, P.O. Box 241, Cape Town 8000, South Africa; 7Flow Cytometry and Immunology Facility, Medical Research Council Unit, The Gambia at London School of Hygiene and Tropical Medicine, Fajara, Banjul P.O. Box 273, The Gambia

**Keywords:** rotavirus, ROTARIX^®^, vaccine, safety, booster dose, immunogenicity, Zambia, Africa

## Abstract

Oral rotavirus vaccines show diminished immunogenicity in low-resource settings where rotavirus burden is highest. This study assessed the safety and immune boosting effect of a third dose of oral ROTARIX^®^ (GlaxoSmithKline) vaccine administered at 9 months of age. A total of 214 infants aged 6 to 12 weeks were randomised to receive two doses of ROTARIX^®^ as per standard schedule with other routine vaccinations or an additional third dose of ROTARIX^®^ administered at 9 months old concomitantly with measles/rubella vaccination. Plasma collected pre-vaccination, 1 month after first- and second-dose vaccination, at 9 months old before receipt of third ROTARIX^®^ dose and/or measles/rubella vaccination, and at 12 months old were assayed for rotavirus-specific IgA (RV-IgA). Geometric mean RV-IgA at 12 months of age and the incidence of clinical adverse events 1 month following administration of the third dose of ROTARIX^®^ among infants in the intervention arm were compared between infants in the two arms. We found no significant difference in RV-IgA titres at 12 months between the two arms. Our findings showed that rotavirus vaccines are immunogenic in Zambian infants but with modest vaccine seroconversion rates in low-income settings. Importantly, however, a third dose of oral ROTARIX^®^ vaccine was shown to be safe when administered concomitantly with measles/rubella vaccine at 9 months of age in Zambia. This speaks to opportunities for enhancing rotavirus vaccine immunity within feasible schedules in the national immunization program.

## 1. Introduction

Diarrhoeal disease is ranked third among the global leading causes of morbidity and mortality in young children, responsible for approximately 1.53 million deaths and contributing to over 80 million disability-adjusted life years, most of which occur within Sub-Saharan African children aged below 5 years [1]. Among several infectious aetiologies of diarrhoeal disease, rotavirus is the most common cause of moderate to severe and less severe diarrhoea [2,3] and the leading cause of diarrhoeal disease-associated mortality that has been attributed to 128,515 deaths in a single year in this population [4].

The orally administered and widely introduced rotavirus vaccines ROTARIX^®^ (GlaxoSmithKline) and RotaTeq^®^ (Merck) have proved important early life interventions in mitigating the diarrhoeal disease burden in this population, with substantial reductions in rotavirus-associated and all-cause diarrhoea morbidity and mortality observed since their introduction [5]. However, vaccine immunogenicity and efficacy when administered in early infancy is consistently lower and variable in highly burdened and high mortality settings with several reasons postulated [6,7]. Improved vaccine performance is an important way in which rotavirus infections that occur even among vaccinated infants can be further prevented. For these oral rotavirus vaccines to provide maximal benefit in these settings, it is important to assess potential methods to enhance the immunogenicity of existing vaccines as their routine use continues.

Removal of the age restrictions for child vaccination and alternate schedules with booster doses of existing oral rotavirus vaccines have the potential to improve vaccine performance with benefits outweighing potential associated intussusception risks [8,9]. Modelling data predicts enhanced anti-rotavirus immunity from booster dose administration at 9 or 12 months of age and prevention of up to 19,600 additional rotavirus-associated deaths in the second year of life annually [10]. Administration of monovalent ROTARIX^®^ and pentavalent RotaTeq^®^ concomitantly with measles vaccine at 9 months has been demonstrated to induce significantly increased anti-rotavirus antibody titres without interference with measles seroresponses in Bangladeshi and Malian infants, respectively [11,12].

Zambia introduced the ROTARIX^®^ vaccine in 2013 and recorded a seroconversion rate of 60.2% [13]. Although a significant decline in rotavirus-attributable childhood diarrhoea has been recorded especially in infants [14], it remains necessary to further reduce residual infection and disease burden. Newer oral rotavirus vaccines have been evaluated in our setting with similar low rates of vaccine seroconversion observed [15]. To date, no study has been conducted on safety and potential immunogenicity benefits of a booster ROTARIX^®^ vaccine dose administered at 9 months of age in Zambian infants.

We aimed to assess a booster dose of ROTARIX^®^ vaccine administered at 9 months of age as an alternative to the current two-dose schedule to enhance anti-rotavirus immunity in Zambian infants.

## 2. Materials and Methods

### 2.1. Study Design and Sample Size Calculation

The study was a single-center, open-label, randomised, controlled trial assessing the safety and immunogenicity of a booster dose of the monovalent ROTARIX^®^ vaccine at 9-month infant age. We anticipated a 15% or greater increase in log10 RV-IgA response after the booster ROTARIX^®^ dose. Using a two-sample *t*-test and assuming equal SD at 5% level of significance, we therefore required a total of 196 infants (98 per arm) to detect an increase to 3.13 log10 RV-IgA due to the booster dose of ROTARIX^®^ at an 80% power. We made an upward sample size adjustment of 9% to account for potential loss to follow-up to reach the total of 214 infants to be recruited in this study. The estimation was performed using Stata 14 MP “power” command (StataCorp™, College Station, TX, USA).

### 2.2. Study Participant Selection and Enrolment

The study enrolled 214 infants aged 6 to 12 weeks from 13th September 2018 to 15th November 2018 at George Health Centre (GHC), a government-run peri-urban health facility serving a high-density, low-income population in Lusaka, Zambia. Mothers presenting with their infants for routine immunization visits were approached by study staff and sensitized about the study. Interested mothers were provided further study information at the clinical research site located within the GHC premises. Mothers that were willing to participate were individually taken through an informed consent process and simple comprehension assessment test in private rooms. Eligibility criteria included that the infant was aged 6 weeks to 12 weeks old; the mother participated voluntarily, provided written informed consent (with a witness in the case of illiterate participant) and agreed to all study procedures; and the mother was resident in the study area and willing to come for scheduled visits for the duration of the study. Infants were not eligible if they had a contraindication to rotavirus vaccination; previously received rotavirus vaccination; had a recent history of immunosuppressive therapy; had a recent history of blood or blood product transfusion; existing congenital anomalies; or any condition deemed by the study investigator to pose potential harm to the participants or jeopardize the validity of study results.

### 2.3. Study Procedures and Randomization

Enrolled mother-infant pairs were followed up until the infant was 36 months old. At baseline, eligible infants were randomised at a ratio of 1:1 using masked allocation into either the intervention arm to receive a booster dose of ROTARIX^®^ concomitantly with measles/rubella (MR) vaccination or into the control arm to receive only MR vaccination at 9 months old. All children in the study received routinely administered first and second ROTARIX^®^ vaccine doses (given from 6 weeks and ideally 4 weeks apart before the age of 2 years). ROTARIX^®^ is an orally administered live, attenuated G1P [8] monovalent vaccine in routine use in Zambia. The batch number of ROTARIX^®^ used in the study was AROLC044AA. Infants in both arms also received polio, Bacillus Calmette–Guérin (BCG), pentavalent diphtheria/pertussis/tetanus/Hepatitis B/*Haemophilus influenza*-type (DPT-HepB-Hib) and pneumococcal conjugate vaccines (PCV) according to the routine Zambian expanded immunization schedule.

Baseline sociodemographic and clinical data were collected from the participating mother/infant pairs. From all enrolled infants, whole blood samples (3–4 mL) were collected before receipt of the first ROTARIX^®^ dose, 1 month after two-dose ROTARIX^®^ vaccination, before receipt of MR vaccine (control arm) or MR and booster dose ROTARIX^®^ at 9 months of age and when 12 months old. In a subset of infants, additional blood sampling was performed within 1 month after the first ROTARIX^®^ dose. From baseline to the time the infant was 36 months old, anthropometric growth measurements were taken and data on incidence of clinical illness were recorded.

### 2.4. Immunogenicity Assessment

Plasma from whole blood samples was tested for anti-rotavirus immunoglobulin A (RV-IgA) titres using an adaptation of a published and validated sandwich enzyme-linked immunosorbent assay (ELISA) based on the use of WC3 rotavirus antigen and mock infected African green monkey kidney (MA104) cell lysate [16]. All plasma testing for RV-IgA was performed at the Centre for Infectious Disease Research in Zambia Enteric Disease and Vaccine Research Laboratory in Lusaka, Zambia. In-house-generated pooled plasma from rotavirus-vaccinated adults was validated for use as the standard in the ELISA assay using pooled serum with known assigned RV-IgA units per millilitre (U/mL). The primary immunogenicity endpoint was the geometric mean titre of anti-rotavirus IgA at 12 months of age. The study also investigated RV-IgA seropositivity and vaccine seroconversion using published definitions. Seropositivity was defined as an RV-IgA titre ≥ 20 U/mL. Seroconversion was defined as a four-fold or greater change in RV-IgA titre 1 month after dose two if pre-vaccination titre was <20 U/mL [13].

### 2.5. Safety Assessment

All enrolled infants received ROTARIX^®^ vaccination together with other routine vaccines as per the Zambian immunization schedule. Prior to vaccination, all participants were screened for any medical condition. Following vaccination, all infants were reviewed by the study staff to identify any immediate adverse events (AE). Participant mothers or guardians were provided with and trained in completing a post-vaccination diary card to record presence or absence of solicited AE including fever, diarrhoea, vomiting, loss of appetite and irritability over the next 5 days following immunization, which was returned to the study clinic at the next study visit. Mothers were also encouraged to bring the infant to the study clinic whenever the child was unwell, at which point standard of care was given and the presenting AE was recorded using structured case report forms. For the AE, information collected included but was not limited to the presenting symptoms, evolution of the presentation of symptoms, examination findings, investigations and drugs given (dosage, route and duration). In the case of serious adverse events (SAE), every effort was made to make physical contact and access the medical records in the admitting health facility. For both AE and SAE, the infants were followed up until resolution whilst offering the necessary standard medical care. Once resolved, the study participant documents were updated accordingly and where required, the local authorities were updated accordingly as per regulatory guidelines. All SAE were also reviewed at regular intervals by the study Data Safety and Monitoring Board (DSMB) comprised of clinicians from the study, those independent of the study and reported to the relevant national regulatory authorities. During routine scheduled study visits to the clinic, mothers were also specifically asked about diarrhoea occurrence and any other illnesses that the infant may have had in the period preceding the visit. All stool samples passively collected from children presenting with diarrhoeal disease during clinic visits were tested for rotavirus. Genotyping was performed on all rotavirus-positive stool samples to determine infecting strains. We documented and described the incidence of clinical AE and SAE within a month following administration of the third dose ROTARIX^®^ + MR and MR alone in infants in the intervention and control arm respectively as the primary measure on safety.

### 2.6. Statistical Analysis

For the immunogenicity analysis, the characteristics of participating infants at 9-month follow-up were tabulated for each arm. Analysis was based on the intention-to-treat population. In the primary analysis, we used two-sample t-test to test the difference in RV-IgA titre at 12-month infant age between the two arms. Linear regression model on log-transformed RV-IgA titre at 12-month infant age was used to estimate the geometric mean ratio and 95% confidence interval (CI), adjusted for potential confounders. *p*-values were considered significant at 5%. For the safety analysis, AE, and SAE incidence within 1 month after receipt of booster ROTARIX^®^ dose and MR vaccine or MR vaccine alone were tabulated for each arm and 95% CI was calculated for the proportion of infants with any AE or SAE in each arm. All analyses were performed in Stata 17 MP (StataCorp, College Station, TX, USA) and R-Software.

## 3. Results

### 3.1. Participant Enrolments and Baseline Characteristics

As summarised in Figure 1, the study enrolled and randomised 214 infants between 13th September 2018 and 15th November 2018. Pre-vaccination whole blood was obtained from 211/214 (98.6%) enrolled infants at baseline. 170/214 (88/170 in intervention and 82/170 in control arm) infants had a clinic visit 28 days after their second dose of ROTARIX^®^. A total of 168 out of 214 (78.5%) infants attended and gave a whole blood sample at their 9-month-age study visit of which 88/168 (52.4%) infants were in the intervention (ROTARIX^®^ + MR vaccination) arm and 80/168 (47.6%) infants in the control (MR vaccination) arm. Of these, 159/168 (94.6.2%), of which 85/159 (53.5%) and 74/159 (46.5%) were in the intervention and control arm, respectively, also attended and gave a whole blood sample at their 12-months-of-age study visit. Infants that had 9- and 12-months-of-age whole blood samples collected were included in the final analysis, whereas others were not included due to dropouts caused by mother’s relocation from study site, withdrawal of consent, non-study related infant deaths, and losses to follow-up of participating mothers during follow-up period.

As outlined in Table 1, infants were from low-income households with poor water sanitation and hygiene (WASH). The majority of the infants were from households with shared toilet facilities and using public water sources. Infants at enrollment had a median age of 6 weeks, the majority were HIV unexposed, full-term with normal weight at birth, generally healthy and mostly breastfed. The RV-IgA seropositivity (RV-IgA titre ≥ 20 U/mL) rate was low at baseline at 4.8% overall and 3.5% and 6.3% in the intervention and control arms, respectively. There was no statistically significant difference in these baseline characteristics between the two study arms.

### 3.2. Seroconversion Rates and Anti-Rotavirus IgA Titres in Two-Dose and Booster Dose ROTARIX^®^ Vaccinated Infants 

As shown in Figure 2, pre-vaccination mean RV-IgA antibody titres were low in the infants but increased after each ROTARIX^®^ vaccine dose. Statistically significant increases in mean RV-IgA titres were observed between baseline and 1 month after the first dose of ROTARIX^®^ in both the control arm (*p* = 0.046) and intervention arm (0.012). However, this increase was less apparent between the first and second doses for both control (*p* = 0.447) and intervention arms (*p* = 0.068). Interestingly, after two-dose vaccination, significant increases in RV-IgA titres in the control (*p* = 0.001) and intervention arms (*p* < 0.001) were observed by 9 months of age. Similarly, a significant increase (*p* < 0.001) in RV-IgA titres was seen in both arms by 12 months of age. 

In general, mean RV-IgA antibody titres were similar in the intervention and control arms at baseline (*p* = 0.06), 1 month after the first dose (*p* = 0.944) and 1 month after the second dose (*p* = 0.644). Similarly, mean RV-IgA titres in the two arms at 9 months old were not significantly different (*p* = 0.207), but the mean RV-IgA titres at 9 months old showed a higher trend among infants in the intervention arm. At 12 months old, the difference in mean RV-IgA titres between the control and intervention arms did not reach statistical significance (*p* = 0.688). 

Vaccine seroconversion approximately 1 month after two-dose ROTARIX^®^ was low in this study population with 47/169 (27.8%) infants seroconverting, of which 25/47 (53.2%) were from the intervention arm and 22/47 (46.8%) from the control arm.

### 3.3. Effect of Booster Dose ROTARIX^®^ at 9 Months on Anti-Rotavirus IgA Geometric Mean Titres at 12 Months of Age 

We observed no statistically significant differences in RV-IgA GMT ratios at 12 months of age between infants that received the third ROTARIX^®^ vaccine dose and those that did not (Table 2). 

### 3.4. Safety: Incidence of Adverse Events and Serious Adverse Events by Trial Arm

Primary safety assessment was conducted on infants who successfully attended the 9-months-of-age study visit and remained in follow-up 1 month thereafter. In these infants, respiratory tract illness (RTI) was the most common AE, followed by diarrhoeal disease with comparable incidence between the intervention and control arms (Table 3). Other AEs observed included conjunctivitis, dermatitis, candidiasis, febrile illness, emesis and otitis with comparable incidences between the two arms (Table 3). Out of 76 stool samples that were passively collected from infants presenting with diarrhoea during unscheduled visits, 4 (5.3%) tested positive for rotavirus. Genotyping of 3 out of the 4 stool samples that had sufficient volumes revealed two G3 and one G4 genotype. Of the G3 genotype infections, one was in an infant in the intervention arm and the other was an infant in the control arm. The G4 genotype was observed in an infant from the control arm.

Throughout the three year study follow-up period, a total of 30 SAEs were recorded. Among these SAEs, 7/30 (23%) had acute gastroenteritis among the presenting symptoms. The study recorded four deaths among these SAE, of which three were in the control arm and one was in the intervention arm.

Only two SAEs, one within each arm, occurred within 1 month after the intervention at 9 months. The SAE recorded in the control arm was acute gastroenteritis with severe dehydration in severe anemia and failure to thrive. The SAE recorded in the intervention arm was acute gastroenteritis with severe dehydration. None of these SAEs recorded were related to the study (Table 4).

## 4. Discussion

In this clinical trial, we assessed the safety and immune boosting effects of a third dose of ROTARIX^®^ vaccine administered at 9 months of age. This is the first clinical trial assessing administration of this oral rotavirus vaccine in Zambia outside of the recommended age range and our data show that a third dose of ROTARIX^®^ given at 9 months of age in Zambian infants is well tolerated. Our results are consistent with studies conducted elsewhere, where no difference in AE and/or SAE frequency was observed between intervention and control arms [11,12].

We found no difference in geometric mean titres and ratios of anti-rotavirus IgA at 12 months of infant age in the intervention arm from a booster third dose of ROTARIX^®^ vaccine given at 9 months compared to the control arm. This contrasts with findings from a study conducted in Mali where a three-fold or greater rise in RV-IgA and greater seropositivity rate 28 days after vaccination was seen among infants who received the booster dose of pentavalent ROTATEQ at 9 to 11 months of age (in addition to doses given at 6, 10 and 14 weeks of age) compared to those who did not. [12]. Another study in Bangladesh also observed an increase in RV-IgA seropositivity and geometric mean titres in infants given a booster dose of ROTARIX^®^ at 9 to 10 months when immunogenicity outcome was assessed 2 months later. This was in comparison to infants that received measles/rubella vaccine alone in which no apparent changes were observed [11]. Both these studies made use of the same WC3 based ELISA methods as used in our current study.

A notable difference of these two studies with our study was that immunogenicity assessment was performed earlier at 1 month and 2 months after rotavirus booster vaccination, whilst our study measured the immunogenicity effect 3 months later. The peaking of RV-IgA tends to occur within 1 month after vaccination, and it is possible that the 3 month period in our study saw a waning of vaccine induced immune responses in the intervention arm such that by our outcome sampling timepoint RV-IgA levels became comparable to the control arm. We chose to assess boosting at 12 months of age as we believed the timepoint was close enough to detect a boosting effect and gave a window between blood sampling timepoints that reduced the frequency of blood draws.

Additionally, of note is the influence that natural rotavirus immunity may have on observed booster dose immunogenicity. The Malian study observed rise in RV-IgA seroresponses among infants who did not receive the additional ROTATEQ dose, suggesting natural rotavirus exposure may have contributed to a rise in titres [12]. We observed similar increase in RV-IgA among infants who did not receive the third dose within the 3 months after intervention. This may indicate that infants in our study had exposure to wild-type infection and the exposure during the three-month period after intervention in our study may have factored into results observed between arms. In Mali, about half of the infants had RV-IgA titres below <20 U/mL (seronegative) prior to receiving the booster dose [12]. In Bangladesh, pre-boost RV-IgA seropositivity was ~52.7%; however, an improvement in boosting effect was observed among infants that were seronegative pre-boost. In our study, higher levels of RV-IgA titres relative to post-two-dose vaccination were apparent in infants at 9 months of age with slightly higher levels in the intervention arm though difference did not reach significance. These higher pre-boost titres in the intervention arm could perhaps have influenced responses observed in diminishing immunogenicity of the booster dose. Nevertheless, differences in population ages, time post-boost and vaccines assessed (monovalent versus pentavalent) could also play roles in these contrasting findings.

This study had the opportunity to investigate pre-vaccination seropositivity and vaccine seroconversion as secondary immune measures. We found minimal baseline rotavirus seropositivity and low post-ROTARIX^®^-vaccination seroconversion rate comparable to estimates reported in a study performed within a similar population in the same setting [15]. These findings show that while ROTARIX^®^ vaccine is immunogenic among infants in our setting, the phenomenon of modest immunogenicity persists. Although our study was not designed to assess the protective effect of vaccination, rotavirus infections were present, and incidence of diarrhoea was among the commonly reported illnesses among vaccinated infants. Detected rotavirus infections were G3 and G4 non-vaccine strains. Whilst ROTARIX^®^ is a monovalent vaccine containing G1P [8] strain protection against non-vaccine infecting strains has been shown [17]. Nevertheless, detection of non-vaccine strains of rotavirus infections among ROTARIX^®^ vaccinated infants may reduce the effectiveness of these vaccines within our settings and speaks towards the need for vaccines covering multiple strains. Such findings in this study emphasize need for continued surveillance of circulating rotavirus strains including other viral, bacterial and parasitic enteric pathogens that may become important in the post-vaccine era.

Among the strengths of the study was the that it was a randomised control design and was conducted in a population in which rotavirus vaccines would be of most benefit. The local implementation of an ELISA method that is widely employed in other rotavirus vaccine trials elsewhere was another strength that enabled comparison of findings to other similar studies. Generally, there are limited studies assessing booster rotavirus vaccine doses at later ages in Africa and this study was the first to be performed in Zambia. Another strength was the ability in our study to demonstrate rotavirus immunity status of the children from pre-vaccination. Our study design enabled determination of pre-vaccination immune status and seroconversion rates after routine two-dose vaccination and accounting for this in our interpretations which was not done in the two studies conducted in Bangladesh and Mali [11,12]. This study design also allowed determination of seroresponses of the vaccine in different localities and sub-population but within the same setting of Zambia by comparison to that performed previously when vaccination was introduced [13].

Notable study limitations included the high losses to follow-up encountered early during the trial which may have reduced the power to detect the boosting effect of the third dose. We measured RV-IgA as an immunogenicity outcome, and, while being the most widely utilised measure for rotavirus vaccine immunogenicity, it is a sub-optimal correlate [18]. Measurement of other rotavirus-specific humoral and cellular immune responses to vaccination is necessary to further inform immunogenicity and potentially correlates of protection. We did not assess the potential impact of the third rotavirus vaccine dose on immunogenicity of the measles/rubella vaccine in our setting; however, studies conducted elsewhere have observed no influence of booster oral rotavirus vaccine given at this age on measles vaccine responses and attainment of sero-protection [11,12].

## 5. Conclusions

Despite showing evidence that ROTARIX^®^ vaccine is well tolerated at 9 months of age, our study findings do not support improved immunogenicity by 12 months of age from a booster dose vaccination at this age in our study setting. However further research is needed to generate stronger clinical evidence for policymakers. Evaluation of alternative vaccine formulations for improved immunogenicity may be important in our setting to increased effectiveness and further reduce the burden of rotavirus.

## Figures and Tables

**Figure 1 vaccines-11-00346-f001:**
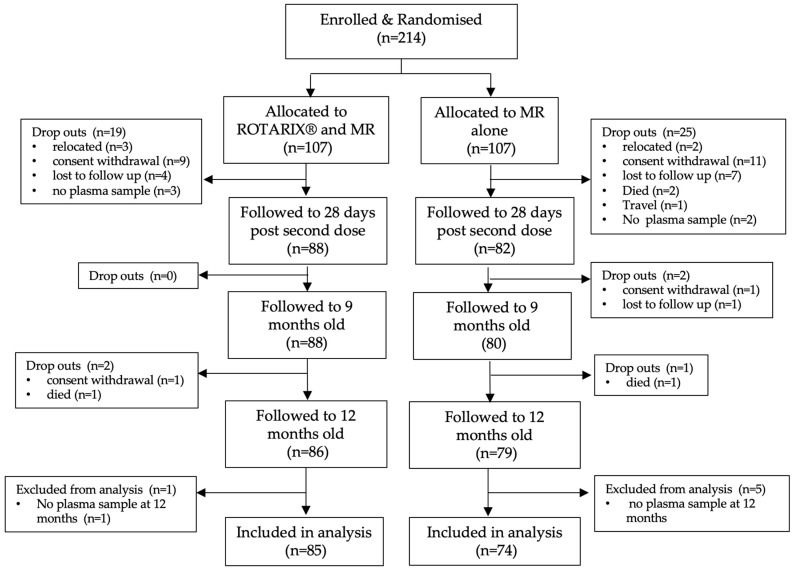
Study participant flow chart. Abbreviations: MR (measles/rubella vaccine).

**Figure 2 vaccines-11-00346-f002:**
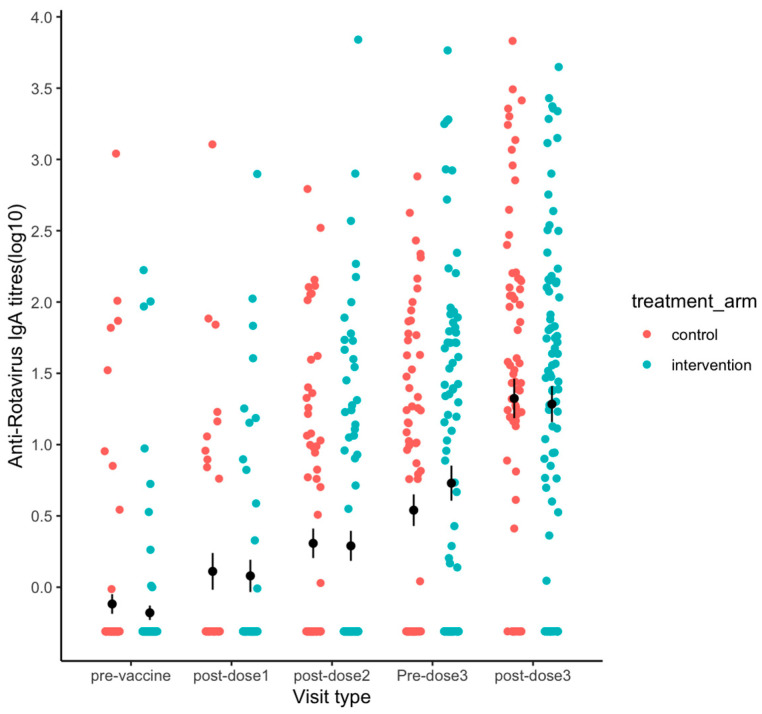
Trends in rotavirus-specific immunoglobulin A (RV-IgA) titres pre and post rotavirus vaccination compared between the control (red circle) and intervention (blue circle) arms. Each circle represents an infant’s log10 RV-IgA titre. Black circle represents mean and standard error of log RV-IgA titre.

**Table 1 vaccines-11-00346-t001:** Baseline characteristics of mother/infant pairs by trial arm.

	Total Population (N = 168 ^a^)	ROTARIX^®^ + MR (n = 88)	MR (n = 80)
	n (%)	n (%)	n (%)
Infant Characteristic			
**Age, weeks**			
Median (IQR)	6 (6–6)	6 (6–6)	6 (6–6)
**Sex**			
Male	89 (53.0)	38 (43.2)	41 (51.3)
Female	79 (47.0)	50 (56.8)	39 (48.8)
**Gestation**			
Pre-term	11 (6.6)	5 (5.7)	6 (7.5)
Full-term	157 (93.5)	83 (94.3)	74 (92.5)
**Mode of Delivery**			
Vaginal	160 (95.2)	84 (95.5)	76 (95.0)
Caesarean	8 (4.8)	4 (4.6)	4 (5.0)
**Feeding**			
Expressed/direct Breastmilk	158 (94.1)	83 (94.3)	75 (93.8)
Mixed breast and formula	10 (6.0)	5 (5.7)	5 (6.3)
**Birth weight, kg (N = 167)**			
<2.5	11 (6.6)	3 (3.5)	8 (10)
≥2.5	156 (93.4)	84 (96.6)	72 (90.0)
**Weight at enrolment, kg**			
Mean (SD)	4.6 (0.6)	4.6 (0.6)	4.7 (0.7)
**Length at enrolment, cm**			
Median mean (SD)	54 (2.6)	54 (2.7)	54 (2.6)
**Malnourished (WLZ < −2) (N = 167)**			
No	164 (98.2)	85 (97.7)	79 (98.8)
Yes	3 (1.8)	2 (2.3)	1 (1.3)
**Stunting (LAZ < −2**)			
No	138 (82.1)	70 (79.6)	68 (85.0)
Yes	30 (17.9)	18 (20.5)	12 (15.0)
**Wasting (WAZ < −2)**			
No	153 (91.1)	79 (89.8)	74 (92.5)
Yes	15 (8.9)	9 (10.2)	6 (7.5)
**HIV exposure**			
Unexposed	119 (70.8)	60 (68.2)	59 (73.8)
Exposed	49 (29.2)	28 (30.8)	21 (26.3)
**RV-IgA seropositive (N = 166)**			
No	158 (95.2)	84 (96.6)	74 (93.7)
Yes	8 (4.8)	3 (3.5)	5 (6.3)
**Maternal characteristics**			
**Age, years**			
<20	23 (13.7)	10 (11.4)	13 (16.3)
20–24	53 (31.6)	29 (33.0)	24 (30.0)
25–29	51 (30.4)	27 (20.7)	24 (30.0)
≥30	41 (24.4)	22 (25.0)	19 (23.8)
**Parity**			
Low parity (1–2)	98 (58.3)	50 (56.8)	48 (60.0)
Multiparity (3–4)	54 (32.1)	27 (30.7)	27 (33.8)
Grand multiparity (5+)	16 (9.5)	11 (12.5)	5 (6.3)
**Education level**			
No education	6 (3.6)	5 (5.7)	1 (1.3)
Some/complete primary	55 (32.7)	29 (33.0)	26 (32.5)
Some/complete secondary	102 (60.7)	52 (59.1)	50 (62.5)
Attended/completed university	5 (3.0)	2 (2.3)	3 (3.8)
**Monthly household income, ZMW**			
<500	64 (38.3)	35 (39.8)	29 (36.7)
500–1000	49 (29.3)	25 (28.4)	24 (30.4)
>1000	54 (32.3)	28 (31.8)	26 (32.9)
**Share toilet facilities**			
No	33 (19.6)	23 (26.1)	10 (12.5)
Yes	135 (80.4)	65 (73.9)	70 (87.5)
**Source of water**			
Public tap/pipe	93 (55.4)	45 (51.1)	48 (60.0)
Piped into house/yard	33 (37.5)	33 (37.5)	26 (32.5)
Yard/public borehole	8 (4.8)	3 (3.4)	5 (6.3)
Protected/unprotected well	8 (4.8)	7 (8.0)	1 (1.3)

^a^ Infants that attended the 9-month visit. Abbreviations: cm (centimeter); HAZ (height-for-age Z-score); HIV (human immunodeficiency virus); IQR (interquartile range); kg (kilogram); MR (measles/rubella vaccine); RV-IgA (rotavirus-specific immunoglobulin A); WAZ (weight-for-age Z-score); WLZ (weight-for-length Z-score); ZMW (Zambian Kwacha).

**Table 2 vaccines-11-00346-t002:** Rotavirus IgA geometric titre mean ratio at 12 months by study arm.

Arm	N (% of Total)	GMT (95% CI)	Two-Sample *t*-Test, *p*-Value	GMT Ratio (95% CI)	*p*-Value	Adjusted GMT Ratio * (95% CI)	*p*-Value
MR	74 (46.5)	3.98 (3.50–4.51)		1	0.689	1	0.223
ROTARIX + MR	85 (53.5)	3.85 (3.41–4.35)	0.688	0.84 (0.35–2.00)		0.61 (0.27–1.35)	

* Adjusted for malnutrition, sex, water source, income, pre-dose three RV-IgA titres using linear regression on log-transformed titres. Abbreviations: MR (measles/rubella vaccine); GMT (geometric mean titre).

**Table 3 vaccines-11-00346-t003:** Incidence of adverse events within 1 month after third dose ROTARIX^®^ (+MR) compared to MR vaccination.

Arm	Diarrhoea (n), Incidence * (95% CI)	RTI (n), Incidence (95% CI)	Conjunctivitis (n), Incidence (95% CI)	Dermatitis (n), Incidence (95% CI)	Candidiasis (n), Incidence (95% CI)	Febrile Illness (n), Incidence (95% CI)	Emesis (n), Incidence (95% CI)	Otitis (n), Incidence (95% CI)
MR	8 3.33 (1.7–6.7)	12 5.0 (2.8–8.8)	1 0.4 (0.05–3.0)	0	1 0.4 (0.06–3.0)	1 0.4(0.05–3.0)	3 0.8 (0.2–3.3)	1 0.4 (0.06–3.0)
ROTARIX + MR	4 2.4 (0.6–4.0)	8 3.0 (1.5–6.1)	2 0.8 (0.2–3.0)	3 1.1(0.2–1.8)	1 0.4 (0.1–2.7)	1 0.4 (0.05–2.7)	1 0.4 (0.05–2.7)	0
Rate ratio (95% CI), *p*-value	1.75 (0.14–1.51), 0.186	1.23 (0.25–1.48), 0.268	1.82 (0.17–20.05), 0.620	-	0.91 (0.06–14.53), 0.946	0.91 (0.06–14.5) 0.946	0.46 (0.04–5.01), 0.509	-

* Incidence per 1000 infant days. Abbreviations: CI (confidence interval); MR (measles/rubella vaccine); RTI (respiratory tract illness).

**Table 4 vaccines-11-00346-t004:** Occurrence of serious adverse events in intervention (ROTARIX^®^ +MR) compared to control (MR) arm.

Arm	At Least One SAE, Incidence * (95% CI)	At Least One Related SAE, Incidence (95% CI)	Deaths
MR	1 0.4 (0.06–3.0)	0	3
ROTARIX + MR	1 0.4 (0.06–2.8)	0	1
Rate ratio, *p*-value	0.94 (0.06–15.0), 0.9633		

Abbreviations: CI (confidence interval); MR (measles/rubella vaccine); SAE (serious adverse event). * Incidence per 1000 infant days within 1 month after third dose ROTARIX^®^.

## Data Availability

The data presented in this study are available on request from the corresponding author. The data are not publicly available due to institutional data policy restrictions.

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
