# Peer review of "Evaluation of ROTARIX® Booster Dose Vaccination at 9 Months for Safety and Enhanced Anti-Rotavirus Immunity in Zambian Children: A Randomised Controlled Trial"

_vaccines, 2023, doi:10.3390/vaccines11020346_

Round 1

Reviewer 1 Report

  • The study assess the immunogenicity of a third dose of oral ROTARIX vaccine. I congratulate to the authors because the manuscript is clear and presented in a well-structured manner.
  • This is an important topic for further study as it may impact on vaccination strategies in this area. The design of this study is appropriate to test the authors hypothesis
  • the methods section contains the details sufficient to reproduce the study
  • the figures/tables/images/schemes are appropriate. It’s they easy to interpret and understand and the data presented is consistently throughout the manuscript
  • The conclusions are consistent with the evidence and arguments presented

Reviewer 2 Report

The submitted paper show the effect of third dose boos of rotarix.

The paper is intriguing and interesting however I have some concerns that would require some observations

1) are the child's under breast feeding? Do the author's have total IGA of studied children? If yes, adjusting Specific rotavirus IgA for total IgA titres do they have any differences?

2) discussion should adbrige focusing with major attention on results

3) conclusion should state about the findings evidences.

Reviewer 3 Report

1. In the statistical analysis, the geometric mean ratio was adjusted using variables from a stepwise regression model. Please briefly introduce the regression model and explain the assumption/purpose of the adjustment. Please include a reference as support if this is a standard analytical procedure.

2. To monitor the immediate adverse reaction to the vaccine, what diagnostic criteria did you follow to identify adverse events?

3. The enrolled candidates' maternal characteristics, especially education and water source quality, are not balanced. What impact would such an imbalance have? Will this weaken the significance of any conclusions? 
